# Transglutaminase 3 crosslinks the secreted gel-forming mucus component Mucin-2 and stabilizes the colonic mucus layer

Jack D. A. Sharpen [1], Brendan Dolan [1], Elisabeth E. L. Nyström[1], George M. H. Birchenough [1], Liisa Arike [1], Beatriz Martinez-Abad [1], Malin E. V. Johansson [1], Gunnar C. Hansson [1] & Christian V. Recktenwald [1✉]

The colonic mucus layer is organized as a two-layered system providing a physical barrier against pathogens and simultaneously harboring the commensal flora. The factors contributing to the organization of this gel network are not well understood. In this study, the impact of transglutaminase activity on this architecture was analyzed. Here, we show that transglutaminase TGM3 is the major transglutaminase-isoform expressed and synthesized in the colon. Furthermore, intrinsic extracellular transglutaminase activity in the secreted mucus was demonstrated in vitro and ex vivo. Absence of this acyl-transferase activity resulted in faster degradation of the major mucus component the MUC2 mucin and changed the biochemical properties of mucus. Finally, TGM3-deficient mice showed an early increased susceptibility to Dextran Sodium Sulfate-induced colitis. Here, we report that natural iso-peptide cross-linking by TGM3 is important for mucus homeostasis and protection of the colon from inflammation, reducing the risk of colitis.

[1] Department of Medical Biochemistry and Cell Biology, University of Gothenburg, SE 405 30 Gothenburg, Sweden. ✉email: christian.recktenwald@medkem.gu.se

The epithelium in the intestinal tract is covered by mucus that provides protection from luminal challenges and bacterial infiltration[1]. Despite the similar proteome composition, the organization of the mucus gel network differs considerably between the small and large intestine[2]. Whereas small intestinal mucus is non-attached, the colonic mucus is a two-layered system with an attached, bacteria-free inner layer and an outer layer harboring the commensal flora[1,3,4]. The molecular mechanisms determining these structural differences are not well understood. The predominant component of mucus is the gel-forming MUC2 mucin which is synthesized by intestinal goblet cells. It has been shown that $Muc2^{-/-}$ mice develop spontaneous colitis, a pre-stage of colitis-associated carcinoma[5,6]. Furthermore, the MUC2 levels in patients suffering from active ulcerative colitis (UC) are decreased when compared to healthy control patients[7].

The human MUC2 monomer consists of 5,130 amino acids organized in three complete and one partial von Willebrand D (vWD) domains in the N-terminal part followed by the first CysD domain and two Proline-, Threonine- and Serine-rich (PTS) sequences that are separated by the second CysD domain[8,9]. The C-terminus harbors a fourth vWD domain, two vWC domains, and the cysteine-knot. During its transport through the endoplasmic reticulum and the Golgi-network MUC2 monomers first form C-terminal dimers and in the later stages of the secretory pathway N-terminal dimers or trimers[10,11]. Furthermore, the PTS sequences become heavily O-glycosylated to form mucin domains. This post-translational modification (PTM) shifts the mass of MUC2 from roughly 650 kDa to more than 2.5 MDa. During the later stages of the secretory pathway isopeptide bonds are introduced probably contributing to the insolubility of MUC2 in chaotropic salts like guanidinium chloride[12]. An enzyme family which can catalyze these natural protein cross-links are the transglutaminases (TGM).

Transglutaminases (R:protein-glutamine γ-glutamyl-transferases; E. C. 2.3.2.13) comprise a family of $Ca^{2+}$-dependent acyl-transferases that can catalyze the transamidation or deamidation of protein-bound glutamine residues which can lead to natural cross-links through the formation of an isopeptide bond between the side chains of glutamine and lysine. This PTM is known to limit protein degradation by conformational changes and modification of protease-labile Lys residues[13,14]. There are nine mammalian TGMs of which TGM2 is the most ubiquitously expressed isoform[13,15]. This isoform is predominantly localized in the cell cytosol but can also be found associated with the plasma membrane. Furthermore, it can be secreted by unknown mechanisms after P2X7 receptor activation[16]. The enzymatic activity of TGM2 is normally silent but during mechanical injury it becomes activated and functions as a wound healing enzyme by stabilizing extracellular matrix (ECM) and cell-ECM interactions[17,18]. Another process where TGMs are important is the morphogenesis of the skin. Here, TGM1, 3 and 5 are involved in the formation of the stratum corneum by cross-linking the envelope precursors such as inloricrin and involucrin[19].

Whether transamidation also has a role in the formation and stabilization of intestinal mucus is currently unknown. Mucus and mucins are stored highly concentrated in the granules of goblet cells and expand 1,000-fold in volume upon secretion. If TGM-catalyzed isopeptide cross-links contribute to mucus homeostasis, this processing has to occur after secretion and expansion.

In this work, we analyze if extracellular TGM activity plays a role in organizing the mucus gel in the colon, especially by increasing its stability. To test this hypothesis the abundance of different TGM isozymes was evaluated and their enzymatic activity determined. We found that the formation of $N^{\varepsilon}$-(-γ-glutamyl)-lysine isopeptide cross-links in colonic mucus was based on extracellular TGM3-intrinsic activity. Furthermore, mice lacking this TGM isoform secrete a more protease-sensitive MUC2 molecule. In addition, $Tgm3^{-/-}$ mice are less protected against dextran sodium sulfate (DSS) induced colitis. Together, our observations indicate that TGM-catalyzed cross-links are important for the stabilization/homeostasis of colonic mucus and its resistance against disease-inducing conditions.

## Results

**Transglutaminase 3 is a dominant cross-linking enzyme in the colon.** Firstly, we determined which transglutaminase isozymes were expressed and synthesized in the colonic epithelium. Mouse colon tissue of wild-type (WT) and $Tgm$ knock-out mice were analyzed for protein abundance by using immunohistochemistry (IHC), mass spectrometry (MS) and gel electrophoresis followed by western blot. As we were interested on the impact of transglutaminases on mucus homeostasis a recently published single-cell transcriptomic study[20] analyzing MUC2-producing goblet cells and non-goblet epithelial cells were mined for the expression profile and protein abundance of the various TGM family members. Analyzing mRNA levels in colonic goblet cells and the remaining epithelial cell populations revealed only transcripts for $Tgm2$ and $Tgm3$ genes (Fig. 1a). Next, the TGM2 and TGM3 protein abundance determined by mass spectrometry (MS) in these two cell fractions was extracted from the single-cell study[20]. This analysis revealed approximately 10-times lower levels of TGM3 in the goblet cells compared to the non-goblet epithelial cells whereas the abundance of TGM2 was two-three orders of magnitude lower than TGM3 in the respective cell population (Fig. 1b). To evaluate the tissue localization of TGM2 and TGM3, immunohistochemical analyses were performed in WT, $Tgm2^{-/-}$ and $Tgm3^{-/-}$ animals together with the UEA1 lectin staining for the highly glycosylated MUC2 mucin. None of the strains reacted with the anti-TGM2 antibody, confirming the low levels of this isoform (Fig. 1c). That this antibody was functional was tested on duodenal tissue sections where a signal for TGM2 was easily observed (Supplementary Fig. S1a). In line with the quantitative data from mRNA expression and protein abundance, both WT and $Tgm2^{-/-}$ animals showed a strong staining for the TGM3 isoenzyme in the epithelium and as expected no signal in $Tgm3^{-/-}$ mice (Fig. 1d).

As TGM3 lacks a signal sequence, we determined if TGM3 could nonetheless be secreted into the mucus. To answer this, gel electrophoresis and western blot analyses for TGM2 and 3 in colonic mucus were performed. Recombinantly expressed TGM2 and cleaved TGM3 were also loaded as positive controls either non-activated or activated by $Ca^{2+}$-preincubation (Fig. 1e). The majority of TGM3 was represented by a band migrating around 75 kDa and a weaker signal migrating at approximately 50 kDa in both WT and $Tgm2^{-/-}$ animals. These two bands represent the zymogenic and active form of the enzyme, respectively. Furthermore, several diffuse, but weak, TGM3 signals migrating between 150 and 250 kDa were detected in the WT and $Tgm2^{-/-}$ strains suggesting the self-multimerization of the enzyme and/or its incorporation into substrate proteins. As similar signals were detected in the activated positive control for TGM3, it is likely that self-multimerization occurs in mucus. In contrast, TGM 2 was not detected in the mucus samples of any mouse strain. Specificity of the antibodies used for the respective isoforms was determined by western blot analyses, the anti-TGM3 antibody showed a cross-reactivity <8% against TGM2 and similarly vice versa (Supplementary Fig. S1b) Together the results show that TGM3 is the predominant transglutaminase in the colonic epithelium and the only isozyme detected in the mucus. Furthermore, its expression in goblet cells suggests that its presence in mucus arises at least partly from active secretion and not only from the shedding of cells.

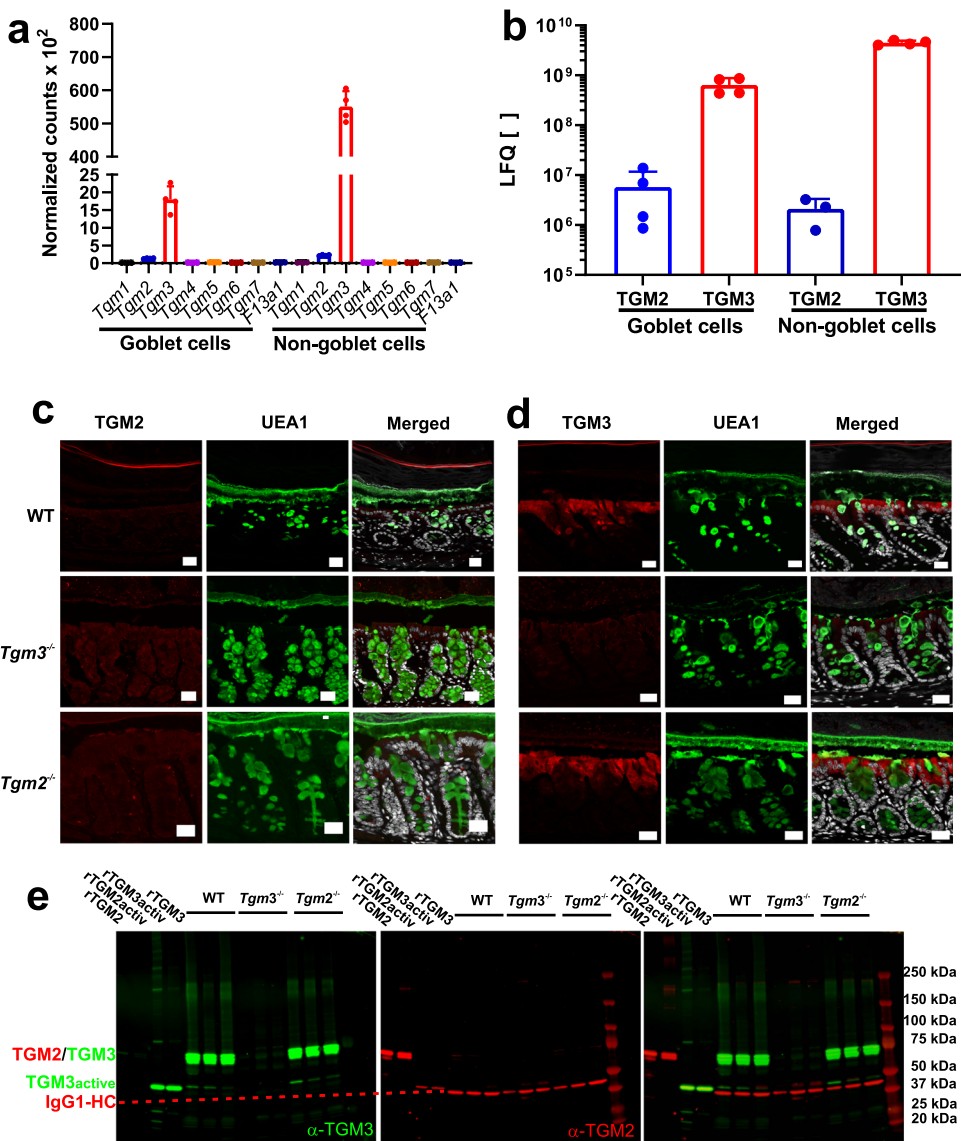

**Fig. 1 mRNA expression, protein abundance and spatial localization of TGM isozymes in the large intestine. a** mRNA-seq expression data of the goblet and non-goblet cell fraction. Goblet cells were separated from other epithelial cell types using FACS-mediated sorting[20]. The graph shows the normalized expression levels of the transglutaminase family members *Tgm1-7* and *F13a1*. $n = 4$. Data are presented as mean ± standard deviation. Source data are provided as a source data file. **b** Label-free quantification of TGM isozymes 2 and 3 in goblet cell and remaining epithelial cells after FACS-mediated cell sorting from RedMUC2[98trTg] mice[20]. The abundance of TGM2 and 3 in the two fractions was measured by mass spectrometry and the data analyzed using the MaxQuant software. $n = 4$. Data are presented as mean ± standard deviation. Source data are provided as a source data file. **c** Immunohistochemistry of colonic tissue specimens from C57/BL6, *Tgm3*[−/−] and *Tgm2*[−/−] mice suggests no TGM2 biosynthesis in the colon. The sections were probed with a monoclonal antibody against TGM2 followed by detection with a secondary antibody coupled to Alexa Fluor 647 (red). The UEA1 lectin (green) was used for goblet cell and mucus visualization. Nuclei were visualized using Sytox green (gray). Scale bar = 20 μm. $n = 3$. **d** Analogously, confocal microscopy of colon specimen from C57/BL6, *Tgm3*[−/−] and *Tgm2*[−/−] mice analyzed for TGM3 (red) using a polyclonal anti-TGM3 antibody that was detected by a secondary antibody coupled to Alexa 647 indicating TGM3 biosynthesis in WT and *Tgm2*[−/−] mice. UEA1 (green) and Sytox green (gray) were used for counterstaining. Images are representative of three biological replicates. Scale bar = 20 μm. **e** Protein abundance analysis of TGM isoforms by Western blot in colonic mucus. The supernatant of precipitated mucus was analyzed for the presence of TGM2 and 3 using a monoclonal anti-TGM2 antibody and a polyclonal anti-TGM3 antibody. Goat anti-mouse IgG1-isoform and anti-rabbit IgGs secondary antibodies were used for visualization on a LI-COR Odyssey Clx workstation. Recombinant non-activated or calcium-activated TGM2 and 3 were loaded as positive controls. The red dashed line marks the IgG1 heavy chain (IgG1-HC) recognized by the secondary antibody against the TGM2 antibody and served as loading control. A representative analysis from three animals per mouse strain is shown.

**TGM3 activity is present in colonic mucus.** Next, we asked if TGM3 is enzymatically active in the colonic mucus and could thereby contribute to its stability by the formation of additional cross-links. For that purpose, a qualitative assay using the incorporation of biotinylated isoform-specific substrate peptides T26 (TGM2) and E51 (TGM3) in mucus was performed. The

mucus was incubated with $Ca^{2+}$ and the respective peptide probe followed by gel electrophoresis and western blot using streptavidin detection (Fig. 2a). Specific incorporation of the two peptides was observed in WT and *Tgm2*[−/−] mucus, but not in mucus from *Tgm3*[−/−] animals. Non-specific signals were observed in all samples, including control reactions where

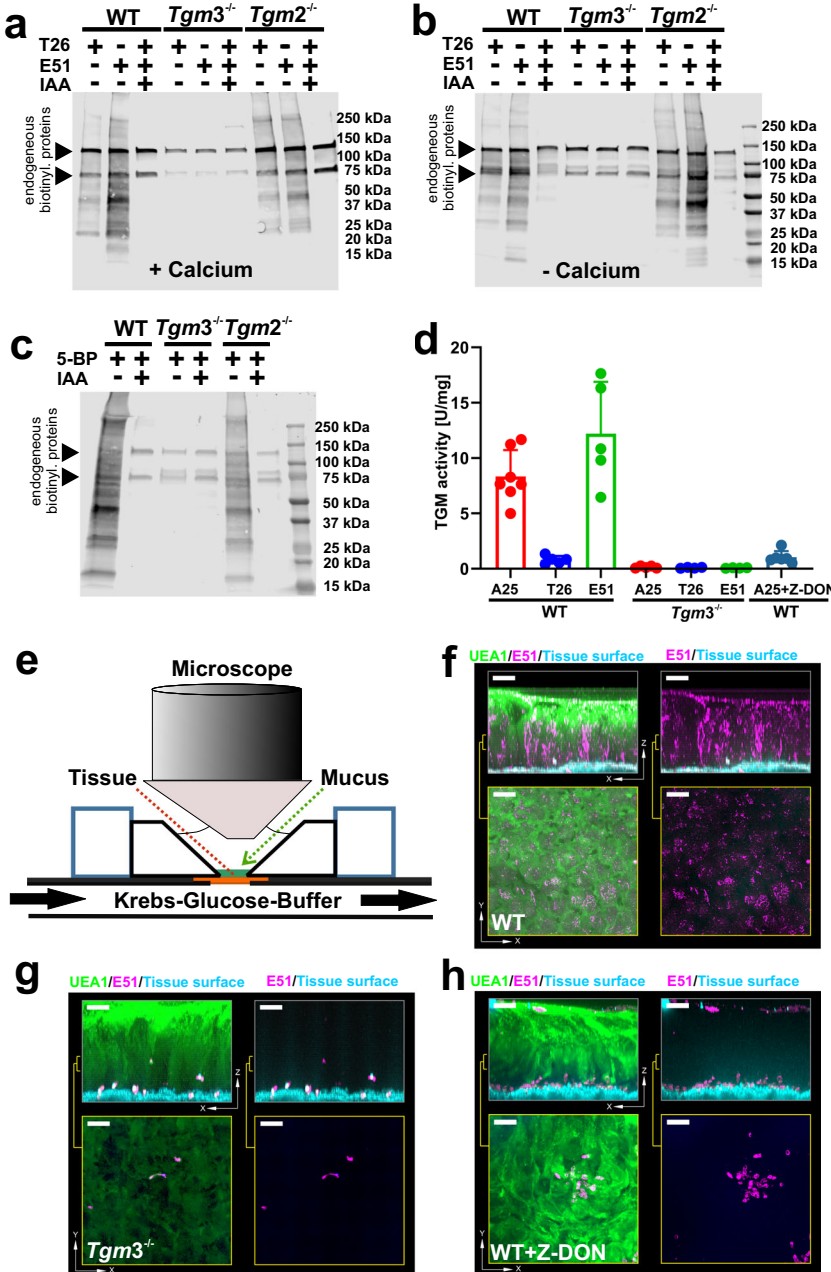

**Fig. 2 Qualitative, quantitative and ex vivo analysis of extracellular transglutaminase activity. a** Qualitative determination of calcium-induced transglutaminase activity in colonic mucus. Samples were treated with biotinylated-TGM2 and TGM3 acyl-donor substrates as described in the methods section. The reaction products were separated by SDS-PAGE and subsequently visualized by Western blot using IR680LT-labeled streptavidin on an infrared imager. Non-specific signals from endogeneously biotinylated proteins were marked with a triangle. A representative example of three biological replicates per mouse strain is shown. **b** Qualitative determination of intrinsic transglutaminase activity in colonic mucus. The samples were treated and analyzed as described in (**a**) without calcium addition. A representative example of three animals per mouse strain is shown. **c** Detection of putative acyl-donor proteins in mucus. Samples from the different mouse strains were incubated in the presence of 5-Bioinyl-pentylamine (5-BP) for 1 h at 37 °C. Control reactions and detection of reaction products were performed as described in (**a**). $n = 3$. **d** Quantitative determination of transglutaminase activity in colonic mucus. TGM activity was determined by the incorporation of TGM2 (T26) and TGM3 (E51)-specific peptides or a promiscuous acyl-donor peptide (A25) into casein as described in the "Methods" section and subsequent normalization against the respective protein concentration. Colored data points mark activities of individual animals ($n \geq 4$). Data present mean ± standard deviation. Source data are provided as a source data file. **e–h** Ex vivo analysis of transglutaminase activity. Tissues were mounted in a perfusion chamber as illustrated (**e**) and transglutaminase activity probed with the glutamine-donor peptide E51 coupled to FITC (magenta) for 30 min at 37 °C. After washing away non-incorporated peptide the tissue specimen were analyzed by confocal microscopy. Mucus and nuclei were counterstained with the UEA1 lectin coupled to rhodamine (green) and the Calcein violet stain (blue) respectively. The top panels show Z-stacks of the explant with (left) or without (right) the UEA1 counterstain. The bottom panels show x/y projections of the indicated area from the respective Z-stack on top. Colonic specimen from WT mice (**f**), $Tgm3^{-/-}$ mice (**g**) or WT mice in the presence of the pan-TGM inhibitor Z-DON (**h**) were probed for E51 incorporation. The scale bar corresponds to 50 µm. Three animals per mouse strain were analyzed.

transglutaminase activity was inhibited by iodoacetamide (IAA). These bands represent likely endogenously biotinylated proteins as for example pyruvate-carboxylase. Thus, the detected cross-linking activity in the mucus arises from TGM3-mediated catalysis. To analyze if endogenous mucus contains sufficient $Ca^{2+}$-ions for the activation of TGM3, the experiment was repeated without calcium addition. Similar results as with exogenous $Ca^{2+}$-addition were obtained, indicating the presence of intrinsic extracellular transglutaminase activity in colonic mucus (Fig. 2b). These results suggest that endogenous acyl-acceptor protein substrates are present in colonic mucus. However, the formation of a transglutaminase-catalyzed cross-linked mucus gel network also requires the presence of acyl-donor proteins. Therefore, the $Ca^{2+}$-free experimental set-up was modified by replacing the glutamine-donor with the primary amine 5-Biotinyl-pentylamine (5-BP) as acyl-acceptor. Similar to the results from the acyl-donor experiments, specific signals were detected when the acyl-acceptor compound was added to mucus of WT and $Tgm2^{-/-}$ animals, but not in the $Tgm3^{-/-}$ mucus or when IAA was added (Fig. 2c). Together, the results show that colonic mucus contains intrinsically, active TGM3 as well as both acyl-acceptor and -donor substrates allowing transamidating reactions to take place.

To quantify the intrinsic transamidating activity in colonic mucus, a colorimetric assay for the incorporation of a TGM-promiscuous peptide (A25) and the two isozyme-selective peptide substrates (peptides T26[21] and E51[22]) into casein was performed (Fig. 2d). A natural cross-linking activity in WT mucus of $\approx 8 \pm 2$ U/mg for the promiscuous substrate was determined. Substitution with the TGM3-specific substrate E51 led to a 1.5-fold increase ($\approx 12 \pm 4$ U/mg) of the transamidating activity, whereas a residual activity of $0.8 \pm 0.3$ U/mg was observed for the TGM2-specific substrate. However, no measurable activity could be obtained in the $Tgm3^{-/-}$ mucus as the detected values were below the limit of detection for our assay (Fig. 2d, Supplementary Fig. S2). Blocking of the TGM-reaction with Z-DON led to an almost complete (88%) inhibition for the promiscuous peptide A25. In line with our other results (Fig. 1, Fig. 2a–c), the natural cross-linking activity was related to TGM3 as the use of the TGM2-specific substrate T26 led to less than 10% transglutaminase activity compared to the TGM3-specific substrate in WT animals and was also below the limit of quantification of this assay. These experiments further demonstrated substantial intrinsic transamidating activity in colonic mucus of WT animals, but not in $Tgm3^{-/-}$, as addition of extra $Ca^{2+}$ was not required.

The intrinsic mucus transamidating activity of TGM3 was further studied using an ex vivo approach where the distal colon from WT and $Tgm3^{-/-}$ animals were mounted in a perfusion chamber and the fluorescently labeled glutamine-donor probe E51 was added and its incorporation monitored (Fig. 2e). Figure 2f–h shows the confocal microscopic analyzes of E51 incorporation in the respective tissue/mucus specimen in the x/z plane (top panels) and snap shots of probe incorporation of the x/y plane inside the mucus (bottom panels). A homogeneous punctuated pattern of E51 fluorescence was observed throughout the whole mucus layers of WT animals (Fig. 2f and Supplementary Movie M1). However, when $Tgm3^{-/-}$ mice were analyzed in the same way, the incorporation was dramatically reduced and limited to shedding epithelial cells (Fig. 2g). A similar lack of incorporation in WT animals was observed in the presence of the transglutaminase inhibitor Z-DON[23] (Fig. 2h). These results demonstrate extracellular TGM3 activity ex vivo. Together these results show that the colonic mucus contains natural acyl-donor and acyl-acceptor substrates together with intrinsic TGM3-mediated transamidating activity.

**Loss of TGM3 alters biochemical properties of mucus/MUC2.** The MUC2 monomer is a large glycoprotein with a mass of around 2.5 MDa (Fig. 3a). It is the most abundant constituent of colonic mucus and is thus a potential target for TGM3-mediated cross-linking, something that could influence its biochemical properties. Colonic mucus from WT, $Tgm2^{-/-}$ and $Tgm3^{-/-}$ mice was isolated and disulfide bonds reduced followed by separation via composite agarose-PAGE (AgPAGE) and detected by in-gel immunostaining using anti-MUC2C3 antibody (Fig. 3b). WT and $Tgm2^{-/-}$ showed two identical diffuse fast-moving bands assumed to be MUC2 monomeric bands (WT-M) and several additional slow-moving and heavily stained bands for higher oligomers. This was in contrast to $Tgm3^{-/-}$ mucus where Muc2 was seen as a faster migrating diffuse band (Fig. 3b; $Tgm3^{-/-}$ M) along with two slower migrating bands similar to WT monomer. These differences in the electrophoretic migration pattern suggest that $Tgm3^{-/-}$ MUC2 is qualitatively different to that of WT and $Tgm2^{-/-}$ and argues for TGM3-mediated isopeptide bond modification of MUC2.

As isopeptide bonds can prevent proteolytic cleavage and secreted mucus is normally exposed to numerous endogenous and bacterial proteolytic enzymes, we hypothesized that the different size of MUC2 formed in $Tgm3^{-/-}$ mice was a result of protease-catalyzed degradation in vivo. To test this hypothesis, colonic mucus of WT and $Tgm3^{-/-}$ mice was first isolated and solubilized by reduction with dithiotreitol. The resulting samples were treated with the serine protease LysC, followed by the separation of the reaction products via composite AgPAGE and Alcian Blue staining of the heavily glycosylated and protease-resistant MUC2 domains (PTS sequence). All three strains showed three identical intensely stained bands after LysC treatment (Fig. 3c). Interestingly, this band pattern was also observed in the non-treated $Tgm3^{-/-}$ mice, but not in the WT or $Tgm2^{-/-}$ animals. This could suggest that the faster migrating MUC2 bands in the non-treated $Tgm3^{-/-}$ animals represent products that have been already degraded in vivo. To confirm this, the fastest MUC2 migrating bands from the non-treated WT (WT-M) and $Tgm3^{-/-}$ ($Tgm3^{-/-}$-M) samples were excised from the gels (Fig. 3b) followed by mass spectrometric analyses of their tryptic/AspN peptides. The peptide coverage of the MUC2 sequence of three biological replicates is summarized in a heat-map shown in Fig. 3d. The WT monomers showed peptides from all domains except the PTS as expected. Interestingly, the $Tgm3^{-/-}$ MUC2 molecule showed almost exclusively peptides from the central CysD2 domain (Fig. 3a, d). The vWD4 domain was weakly covered in both mouse strains explaining the anti-MUC2C3 staining (Fig. 3b). As the fastest migrating bands in the $Tgm3^{-/-}$ mucus were stained by Alcian Blue and have masses larger than 460 kDa, these bands must also include the two mucin domains surrounding CysD2. These PTS1 and PTS2 sequences are highly glycosylated, resistant to proteolytic enzymes, and not identifiable by mass spectrometry (Fig. 3a). Thus, the MUC2 mucin in the $Tgm3^{-/-}$ mice is suggested to be already degraded in vivo due to it being more susceptible to degradation in the colon lumen. In order to explore if bacterial or host proteases are responsible for this degradation of MUC2 WT and $Tgm3^{-/-}$ mice were cohoused and an antibiotic cocktail was supplied via the drinking water for four days. This short treatment led to an approximately 100-fold reduction of bacterial load as determined by the 16S rDNA content in the fecal pellet (Supplementary Fig. S3a). After the treatment the electrophoretic migration pattern of MUC2 was analyzed. Under these conditions no difference between WT and $Tgm3^{-/-}$ animals could be detected (Supplementary Fig. S3b). This data suggests that the degradation of MUC2 in TGM3-deficient mice is mainly driven by the microbial proteases.

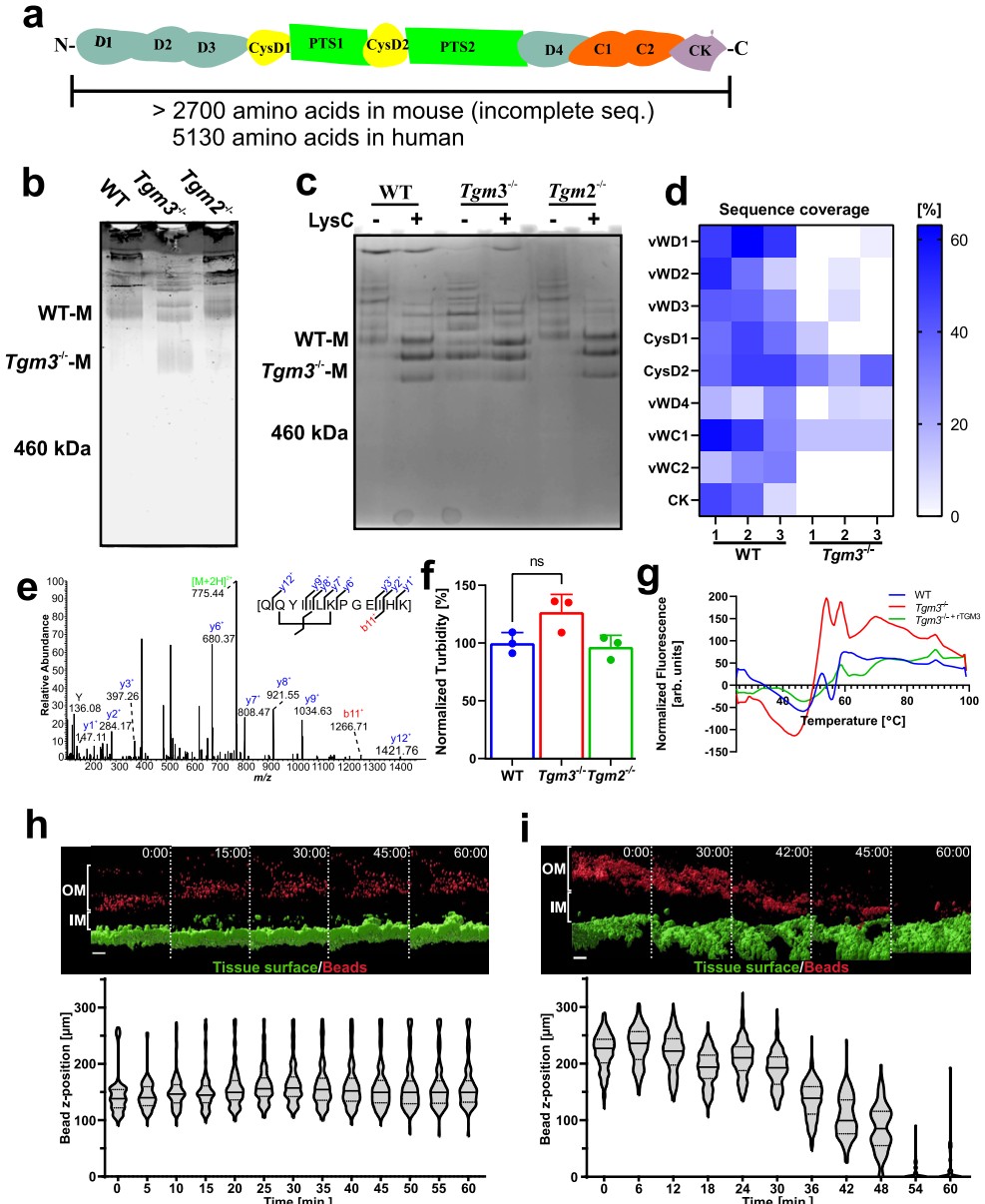

**Fig. 3 Loss of TGM3 causes biochemical alterations of MUC2 and mucus. a** Schematic figure of human and mouse MUC2. Protein domains excluding the signal sequence are shown. Abbreviations correspond to vWD, <u>v</u>on-<u>W</u>illebrand <u>D</u>; CysD, <u>c</u>ysteine-rich <u>d</u>omain; PTS, <u>P</u>roline-<u>S</u>erine-<u>T</u>hreonine rich becomes mucin domain after O-glycosylation; vWC, <u>v</u>on-<u>W</u>illebrand <u>C</u>; CK, <u>C</u>ystine <u>K</u>not. **b** MUC2 mono- and oligomers from WT, $Tgm2^{-/-}$ and $Tgm3^{-/-}$ colonic mucus separated by composite AgPAGE and stained by in-gel immunodetection ($n = 3$). **c** Limited proteolysis of MUC2 by Lys-C. Mucus was incubated in the absence or presence of Lys-C followed by composite AgPAGE analysis and staining with Alcian blue ($n = 3$). **d** MUC2 sequence coverage. MUC2 domains are organized from the N- to C-terminus on the ordinate. The PTS domains were excluded as they cannot be analyzed due to their glycosylation heterogeneity. Non-treated MUC2 monomers (WT-M and $Tgm3^{-/-}$-M) as indicated in (**c**) were analyzed by MS and sequence coverage color-coded ($n = 3$). Source data are provided as a source data file. **e** Detection of transglutaminase reaction products. Example of an intramolecular isopeptide cross-link solely detected in MUC2 from WT animals. MS2 fragment spectrum of the parent ion $[M + 2H]^{2+}$ 775.44 is shown. B-ions are labeled red and y-ions blue. The parent ion is labeled green. **f** Analysis of MUC2 polymerization. Following precipitation of insoluble MUC2 by centrifugation the absorbance of soluble material was recorded at 600 nm. ($n = 3$). Data are presented as mean ± standard deviation. A two-sided t-test between WT and $Tgm3^{-/-}$ samples was performed $p = 0.0738$. **g** Analysis of mucus hydrophobicity. Melting curves were recorded as described in the methods section. Graph shows the mean of three animals per strain. **h**, **i** Pronase treatment of colon explant from WT (**h**) and $Tgm3^{-/-}$ (**i**) mice. Explants were mounted in perfusion chamber (Fig. 2e) and treated with pronase. Mucus surface was visualized by addition of fluorescently labeled beads, epithelium was counterstained with Syto9. The upper panels show isosurfaces of tissue (green) and beads (red) over time. Lower panel shows the distribution of beads in relation to tissue over time as violin plot. Black bars represent median bead position. Dashed lines represent the 25% quartile. $n = 3$. (Scale bars = 50 µm; OM = <u>o</u>uter <u>m</u>ucus; IM = <u>i</u>nner <u>m</u>ucus).

The most likely explanation for the more degraded MUC2 in *Tgm3*[−/−] mice is the loss of protective transglutaminase-catalyzed isopeptide bonds. To search for such bonds, we mined the mass spectrometry data sets for the presence or absence of such cross-links. An example of an intramolecular cross-link connecting Gln 2503 with Lys 2508 is shown in the mass spectrum (Fig. 3e). This intramolecular cross-linked peptide was only detected in MUC2 from WT, but not in *Tgm3*[−/−] animals. This isopeptide bridge is located between the vWC2 domain and the cysteine-knot (CK) domain (Fig. 3a). There are likely several additional cross-links and this isopeptide-bridged peptide is only one example, but its absence in *Tgm3*[−/−] MUC2 supports this interpretation.

As non-reduced secreted MUC2 polymers in the intestine are known to be insoluble in guanidinium chloride due to isopeptide bonds formed intracellularly[24], we asked if TGM3-mediated isopeptide cross-links contributed to this property. To address this question, insoluble mucus from WT and *Tgm3*[−/−] mice was precipitated by centrifugation and the turbidity of soluble material in the supernatant was recorded (Fig. 3f). The turbidity of the samples from *Tgm3*[−/−] animals was increased by approximately 30% when compared to WT and *Tgm2*[−/−] strains. This result further supports the idea that disintegration of the MUC2 mucin network was more prominent in the mice lacking the TGM3 enzyme.

Mucins have been shown to attach to hydrophobic surfaces[25]. We hypothesized that natural isopeptide cross-links might contribute to this biophysical property and thus analyzed the hydrophobic character of colonic mucus by using a thermal fluorescent shift assay. Colonic mucus mixed with the hydrophobic dye SyproOrange was subjected to a linear temperature gradient and the fluorescence measured (Fig. 3g). At higher temperatures (>50 °C) the *Tgm3*[−/−] mucus showed an increased fluorescence in relation to WT, indicating an increased exposure of hydrophobic protein parts. Preincubation of *Tgm3*[−/−] mucus with recombinant TGM3 partly normalized the mucus. It can be suggested that TGM3-mediated isopeptide bonds in WT mucus prevented the unfolding of its constituents.

Mucus processing and tissue secretory responses were assessed using ex vivo mucus measurement assays. Using this approach, we detected no differences in baseline mucus growth rate or carbachol-induced secretory responses between WT and *Tgm3*[−/−] tissues (Supplementary Fig. S4a). A similar approach can be used to measure mucus barrier function by applying bacteria-sized (1 μm diameter) beads to the mucus surface and determining the extent of bead penetration into the mucus via confocal microscopy. However, again no difference between WT and *Tgm3*[−/−] tissues was detected using this approach (Supplementary. Fig. S4b), which was surprising, as we had observed a more degraded MUC2 mucin in the *Tgm3*[−/−] animals. Nonetheless, we hypothesized that lack of TGM3 would affect mucus barrier stability and thus treated colonic tissue from WT and *Tgm3*[−/−] animals with pronase. In WT animals and before addition of pronase to *Tgm3*[−/−] tissue, the fluorescent beads remained on top of the mucus layer (Fig. 3h and Supplementary Movies M2 and M3). However, after pronase treatment of *Tgm3*[−/−] explants, a progressive decrease in mucus thickness was observed and the beads were more easily washed away and/or penetrated down to the epithelial surface indicating that *Tgm3*[−/−] mucus was less protected against proteolytic attack (Fig. 3i and Supplementary Movie M3).

The observed defects of *Tgm3*[−/−]-mucus suggest that its normal barrier function is compromised and that the normal organization of an inner, attached layer devoid of bacteria and the outer layer harboring the commensal flora is disturbed. This hypothesis was addressed by a combined fluorescent in situ hybridisation (FISH)/MUC2 staining in these mice. Whereas tissue sections from WT animals showed a stratified organized inner mucus layer, the mucus of *Tgm3*[−/−] mice had a more disrupted appearance and seemed to be more detached from the epithelium. Furthermore, the separation distance between bacteria and the tissue surface was decreased and the micro-organisms were sometimes in close contact with the epithelium (Supplementary Fig. S5).

**Tgm3[−/−] mice are more susceptible to early DSS-induced damage.** The altered biochemical properties of mucus and its higher susceptibility to proteolytic degradation in the absence of TGM3 activity suggested that *Tgm3*[−/−] mice could be more susceptible to dextran sodium sulfate (DSS) induced colitis. To test this, sex- and age-matched cohoused *Tgm3*[−/−] and WT animals were challenged with DSS. The body weight of WT mice increased during the first four days whereas the *Tgm3*[−/−] animals started to lose weight from day three and showed on trend decreased body weights compared to WT mice until day 6 (Fig. 4a). This was reflected by an earlier detection of occult blood in the feces of *Tgm3*[−/−] mice one day after the start of the experiment (Fig. 4b). Consequently, the *Tgm3*[−/−] animals showed a significant raised disease activity index score (DAI) from days two through five after the start of the DSS treatment (Fig. 4c). Higher DAI was maintained in the *Tgm3*[−/−] compared to WT animals until day 6, when the colitis also became established in the WT animals. Finally, 50% of the *Tgm3*[−/−] animals had to be sacrificed at day 6 or 7, compared to 10% of WT mice, due to suffering and loss of weight following the ethical permit (Fig. 4d). Furthermore, the colon length of *Tgm3*[−/−] mice was reduced to 88% of the WT length after 7 days of DSS treatment (Fig. 4e, f). Histopathological analysis of the colonic tissue after eight days of DSS treatment revealed the loss of crypts and an extensive infiltration of immune cells in both strains. These effects were more pronounced in the distal colon (Fig. 4g). However, histological examination of Hematoxylin/Eosin-stained tissue by a blinded pathologist did not detect significant differences between the two animal strains at the end of DSS treatment. DSS has previously been shown to disrupt the mucus layer properties[26] and mice lacking the MUC2 mucin are very susceptible to disease after only one day of DSS treatment[4]. The early onset of DSS effects in the *Tgm3*[−/−] supports the conclusion that the colonic mucus is defective in these animals. When colonic tissue was analyzed by immunohistochemistry for TGM2, this isozyme that was absent in non-treated WT and *Tgm3*[−/−] as shown in Fig. 1a, was now detected in both the WT and *Tgm3*[−/−] animals after 7 days of DSS treatment (Fig. 4h). Taken together *Tgm3*[−/−] animals were significantly more susceptible towards the colitis-inducing effects of DSS as a faster disease onset was observed resulting in a decreased probability of survival.

## Discussion

We have previously shown that the reduction-insensitive MUC2 oligomers formed in a cell line producing MUC2 were cross-linked by isopeptide bonds as catalyzed by a yet unidentified transglutaminase[12]. However, this colorectal cell line does not secrete MUC2 and could not be used to learn if and how extracellular cross-linking could contribute to mucus homeostasis and colon barrier function. By using WT and knock-out mouse strains, we have uncovered intrinsic transglutaminase activity in secreted colonic mucus mediated by TGM3. The observations provide evidence for the protective effect by natural cross-links. That TGM3 is the dominant transglutaminase of the colon is in accordance with a previous mucus proteome study[2]. mRNAseq and MS studies detected minor amounts of TGM2 but based on

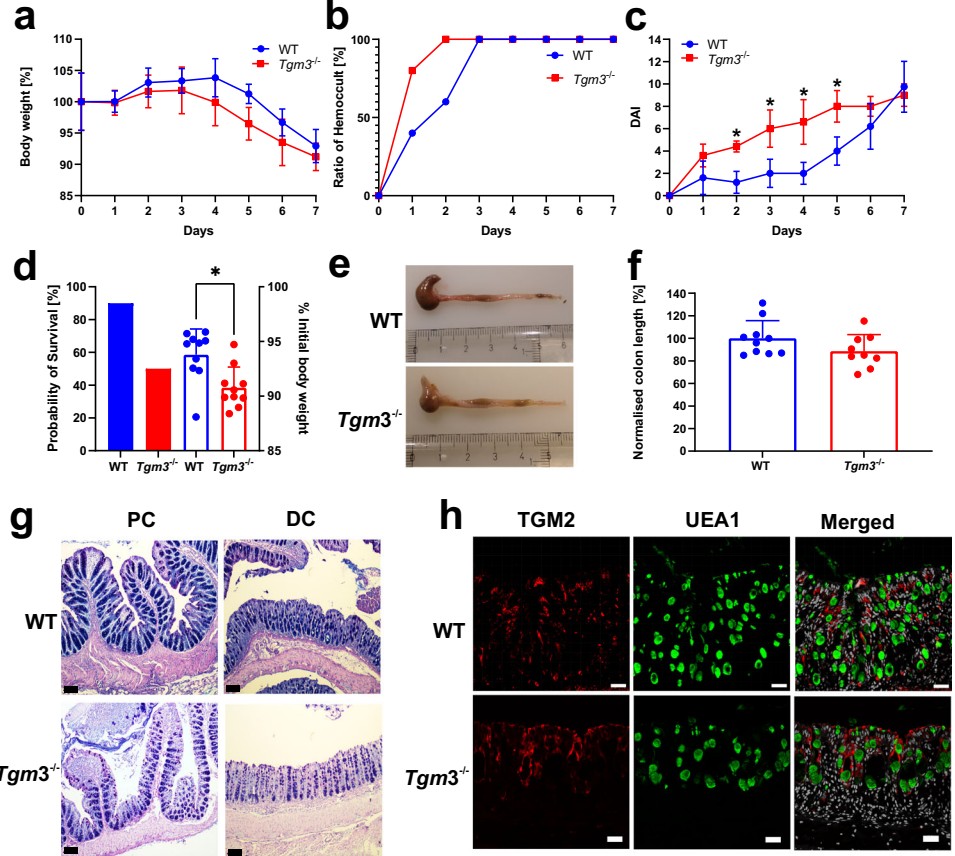

**Fig. 4 Dextran sodium sulfate (DSS) treatment shows decreased mucus protection.** WT and $Tgm3^{-/-}$ mice were cohoused and supplied via drinking water with DSS. Data are presented as mean ± standard deviation. Source data are provided as a source data file. **a** Body weight change of DSS-treated mice over time ($n = 5$). **b** Detection of occult blood. Fecal samples were analyzed for hidden blood using a hemoccult kit. The mean ratio of hemoccult-positive samples from each group was plotted against time ($n = 5$). **c** The disease activity index (DAI) was determined as described in the methods section and the mean with standard deviation for both groups plotted against the time. (Multiple unpaired, two-tailed t-tests). Stars indicate an adjusted $p$ value <0.05 for the respective data point ($n = 5$). **d** Survival analysis of DSS-treated mice. Mice were sacrificed when the initial body weight loss exceeded 10%. The graph summarizes the probability of survival of the two mouse strains (filled bars) and the percentage of the initial body weight at the ethical endpoint (scatter plot; two-tailed Wilcoxon test, $*p = 0.0137$). Two litters per strain were analyzed independently representing ten animals in each group. **e**–**f** Colon length changes of DSS-treated WT and $Tgm3^{-/-}$ mice. At the ethical endpoint, animals were sacrificed and the colon length of each animal measured. A representative colon of WT and $Tgm3^{-/-}$ animals is shown in (**e**). **f** Normalized colon length of DSS-treated WT and $Tgm3^{-/-}$ mice. The graph shows the summary of two litters per strain analyzed independently representing ten animals in the WT and nine animals in the $Tgm3^{-/-}$ group ($p = 0.1214$). **g** Histological analysis of DSS-treated WT and $Tgm3^{-/-}$ mice. Representative Alcian Blue-Periodic Acid Schiff-stained sections from proximal (PC) and distal colon (DC) of WT and TGM3-deficient animals are shown ($n = 5$). The black scale bar corresponds to 100 μm. **h** Immunohistochemical analysis of WT and $Tgm3^{-/-}$ mice for the presence of TGM2 after DSS treatment. Tissue specimen from WT and $Tgm3^{-/-}$ animals were probed with an anti-TGM2 antibody (red) and the UEA1 lectin (green). Nuclei were stained using the Hoechst stain (gray). $n = 5$. The white scale bar corresponds to 30 μm.

the label-free mass spectrometric quantification TGM2 was <1% of that of TGM3 and could represent contaminating material from the ileum. In support of this, TGM2 was not detected by immunohistochemistry or gel electrophoresis/western blot. Previous work from Jeong and coworkers has claimed[27] TGM2 as the major transglutaminase of the colon. However, these authors used only immunohistochemistry to demonstrate the presence of TGM2 and no antibody staining against TGM3 was tested. Likely cross-reactivity of the used antibody can explain this observation.

The strong TGM3 signals observed by gel electrophoresis/ western blot analyses of mucus represented the zymogenic form of TGM3. In addition to this, several TGM3 bands with higher molecular masses were detected in the range between 150 and 250 kDa. Together with control reactions performed with recombinantly activated TGM3, these signals strongly suggest that the enzyme can self-multimerize and/or incorporate itself into other molecules as previously observed for TGM2[28].

Recent reports have shown that TGM2 is extracellularly inactive and can be activated after injury or stress[17,29]. Here, we were able to demonstrate intrinsic, extracellular transglutaminase activity in both the WT and $Tgm2^{-/-}$ mouse strains, but its absence in $Tgm3^{-/-}$ animals. In addition, the obtained information showed the presence of natural acyl-donor and –acceptor substrates in colonic mucus thereby implying the possibility of in vivo isopeptide-based cross-linking of different mucus components. Furthermore, transglutaminase activity could be detected without the addition of calcium, showing that extracellular transglutaminase activity is intrinsic to the large intestinal mucus. Since a > 90% reduction in TGM activity was observed when the TGM2-selective substrate T26 was used in a quantitative assay and no activity was found in the $Tgm3^{-/-}$ animals, we can conclude that the transamidation activity of mucus is almost exclusively dependent on TGM3 in colon. This is in line with the shown absence of TGM2 protein. In an ex vivo assay, a punctated

incorporation of the specific TGM3 peptide substrate E51 in colonic mucus was observed confirming our in vitro observations. However, it was not possible to perform the ex vivo mucus incorporation approach under $Ca^{2+}$-free conditions since the normal cellular signaling of the tissue depends on an extracellular calcium pool. This is reflected by measurements of the luminal calcium concentrations in the gut which vary between 5 and 20 mM depending on the feed state[30,31] whereas the concentration in the used buffer was 1.3 mM representing the physiological luminal calcium concentration[32]. Given that only 20% of the daily calcium intake is resorbed, mainly in the small intestine, the luminal $Ca^{2+}$-concentration in the colon should be sufficient to occupy the second and third $Ca^{2+}$-binding site of TGM3 necessary for its activation[33,34]. Furthermore, TGM3 is expressed and synthesized in goblet cells, a secretory cell lineage whose secretory granules contain high calcium concentrations for the packing of the MUC2 mucin. Secretory granules can contain calcium concentrations of up to 40 mM[35]. The pH in goblet cell granule is acidic and the $Ca^{2+}$-ions are bound to MUC2 and the other stored molecules. After secretion, the pH raises, and free $Ca^{2+}$-ions will become available. In addition to goblet cells, neighboring enterocytes could also contribute to the TGM3-mediated mucus stabilization after their shedding and subsequent release of their cellular content as this acyl-transferase was also abundant in these cells. For the activation of TGM3, the $Ca^{2+}$-binding sites must be occupied and the zymogenic form of TGM3 needs to be cleaved in the loop harboring amino acids 462-469. For this to take place, Cathepsin L or S have been suggested as activating proteases[36]. Interestingly, Cathepsin S is a core mucus component and Cathepsin L is also expressed in colonic epithelial and goblet cells[7] (Supplementary Fig. S6). This suggests that TGM3 can become fully activated in the colonic mucus and lumen, in line with the endogenously observed TGM3 activity in colon mucus. Overall, the availability of calcium and Cathepsin S and L together with an alkaline pH in the large intestinal lumen provide favorable conditions for TGM3 to catalyze transamidating reactions in colonic mucus.

The observed intrinsic, extracellular transglutaminase activity led us speculate about the putative functional impact of the natural cross-links in colonic mucus. The comparison of mucus from WT and $Tgm3^{-/-}$ mice provided direct experimental evidence that the loss of TGM3 led to important biochemical alterations of the dominant mucus skeleton protein MUC2. We observed an extensive degradation of the polypeptide as the N-terminal part with the first three vWD and the first CysD domain was lost comprising approximately 1,300 amino acids as well as most of the C-terminus. Thus, leaving the central part of the MUC2 mucin consisting of the two highly glycosylated PTS sequences linked via the CysD2 domain that is located between them, intact (Fig. 3a). The resistance of the mucin domains to proteolytic cleavage is due to their dense decoration with O-linked glycans resulting in steric hindrance to protease degradation[37]. Another feature of MUC2 that was affected in $Tgm3^{-/-}$ mice is its solubility. MUC2 polymers become insoluble during their transport through the later stages of the secretory pathway[24]. In our turbidity assay, an effect on the MUC2 gel network was indicated by an increased optical density in the mucus supernatant from $Tgm3^{-/-}$ mice. This suggests more soluble MUC2 in this mouse strain. In addition to its solubility, the hydrophobicity of MUC2 was also altered in the absence of TGM3. The assay showed an increased exposure of hydrophobic patches in partly purified MUC2 from $Tgm3^{-/-}$ animals compared to WT animals upon heat-induced denaturation. This may reflect that the $N^{\varepsilon}(-\gamma$-glutamyl)-lysine bonds in WT-MUC2 stabilize the protein. This phenomenon was partially reversed by preincubating the $Tgm3^{-/-}$ mucus with recombinant TGM3, supporting the impact of isopeptide bonds on this

biophysical parameter. Our observations are consistent with previous studies showing that isopeptide bonds can stabilize bacterial pili proteins in this kind of assay[38,39]. However, the insoluble nature of the MUC2 mucin makes it impossible to purify the MUC2 polypeptide in its native conformation, making recording of a specific melting temperature impossible.

In the $Tgm3^{-/-}$ mice, the shortened and more degraded MUC2 still seems to be sufficient to provide enough protection for the colonic epithelium. This mouse strain behaves normally and shows no obvious signs of colon inflammation under normal conditions. It is likely that the highly O-glycosylated mucin domains of MUC2 are sufficient to trap microorganisms and prevent bacteria from reaching the epithelial cells. However, challenging the system by the addition of a mixture of serine proteases deciphered an altered phenotype in $Tgm3^{-/-}$ animals in an ex vivo bead penetration assay. In this approach, the mucus layer seemed to be more disintegrated and less organized allowing the bacteria-mimicking beads to reach the epithelium suggesting a compromised barrier function. We also obtained direct experimental support of this hypothesis by administering DSS to WT and $Tgm3^{-/-}$ animals. DSS quickly disintegrates the inner colon mucus layer allowing bacteria to reach the epithelium and trigger inflammation typically observed after five days[26]. DSS treatment showed that the $Tgm3^{-/-}$ mice were more susceptible and already displayed defects after only two days of treatment. The early onset suggests direct effects on mucus that are likely explained by the decreased intermolecular cross-links in the mucus of $Tgm3^{-/-}$ mice. Less cross-links will, as shown here, make the mucus more susceptible to proteolytic degradation and detachment leading to a faster mucus removal by intestinal peristalsis. Interestingly, two transcriptomic studies using either the 2,4,6-trinitrobenzene sulfonic acid or the adoptive T-cell transfer colitis model detected TGM3 downregulation after the establishment of the disease, thereby suggesting an impact of this enzyme for a healthy gut[40,41]. In contrast to non-treated mice, both the WT and $Tgm3^{-/-}$ mice synthesized the TGM2 enzyme in their colonic mucosa after DSS treatment. This might reflect a role for TGM2 in wound healing as suggested previously[18,27]. As mucus can be regarded as our 'inner skin' it is not surprising that a weakened mucus barrier in the $Tgm3^{-/-}$ mice could be regarded in analogy with the TGM3 function in the skin where earlier observations have shown that these animals have an impaired skin barrier[42].

This study identifies TGM3 as an important natural cross-linking enzyme acting on the expanded secreted mucus and by this contributing to the stabilization of the colonic mucus gel network. The MUC2 mucin and other mucus components are secreted into the harsh luminal environment where proteases from the host, the commensal bacteria, and eventually from pathogens reside. The TGM3-catalyzed formation of isopeptide bond cross-links strengthens the mucus barrier and thereby increase the mucus protection of the colonic epithelium. However, further studies are required to more precisely understand the molecular details of the role of transglutaminases for the mucus structure. For example, as there exists an inverse gradient of TGM2 and 3 abundance from the small to the large intestine it would be interesting to determine the activity of TGM2 and decipher its role for small intestinal mucus. Our observations increase our understanding of the molecular mechanisms that contribute to the architecture of the colonic mucus layers and suggest potential treatment options for the human disease UC.

## Methods

**Animals**. This study complied with all relevant ethical regulations for animal testing and research. All procedures were approved by the Swedish Board of Agriculture (Jordbruksverket) that answers to the Ministry of Agriculture. The used

ethical permits are 2285/19 and 2292/19. C57/BL6N mice were from Taconic. $Tgm2^{-/-}$ mice[43] were provided from Oslo University Hospital (Norway). $Tgm3^{-/-}$ mice[44] were obtained from the University of Rome (Tor Vergata, Italy). Mice were maintained at 22 °C with light/dark cycles of 12 h each at a humidity of 40–70%. Animals received a standard rodent diet and water was supplied *ad libitum*.

**Antibodies, enzymes, chemicals**. If not otherwise specified chemicals were bought from Sigma. For the detection of TGM2 the monoclonal CUB7402 antibody (Thermo Fisher Scientific) was used for both immunohistochemistry (IHC) and Western Blot. TGM3 detection was performed using the polyclonal NBP1-57678 antibody (Novus Biologicals) for both applications. Cross-reactivity of the two antibodies was analyzed by western blot against recombinant TGM2 and 3 (Supplementary Fig. S1b). For IHC detection of TGM2 a goat-α-mouse-IgG1 antibody coupled to AlexaFluor647 (Invitrogen) and a goat-α-rabbit-IgG antibody coupled to AlexaFluor647 (Invitrogen) for TGM3 detection was used. For Western Blot detection of TGM2 and TGM3 a goat-α-mouse-IgG1 antibody coupled to the IRdye 680LT (LI-COR) and a goat-α-rabbit-IgG antibody coupled to AlexaFluor 790 (Invitrogen) were used respectively. The antibody for in-gel immunodetection of MUC2 was from Genetex (GTX100664). Trypsin and AspN were from Promega. LysC was from WAKO (Japan). Recombinant TGM2 (T022) and TGM3 (T013) as well as the biotinylated glutamine-donor substrates A25 (B001); T26 (B008); E51 (B009) and the biotinylated amine-donor compound pentylamine (B002) were bought from Zedira (Germany). The FITC-labeled E51 probe was bought from CovalAb (France). The transglutaminase inhibitor Z-DON[23] was from Zedira. Pronase was from Merck (Germany). The UEA1 lectin was from Vector Laboratories (CA). Dextran sodium sulfate was from TdB consultancy (Sweden). The oligonucleotides for the quantification of the 16S rDNA were synthesized by Eurofins Genomics and had the following sequences: forward primer: 5′-AAACTCAAAKGAATTGACGG-3′ and reverve primer: 5′-CTCACRRCAC-GAGCTGAC-3′. The EUB338 probe for the fluorescence in situ hybridization (FISH) was synthesized from Eurofins Genomics and had the following sequence: 5′-ATTAGTCCATGTTTCCAT-3′ and labeled with AlexaFluor 555.

**Immunohistochemistry**. Paraffin-embedded tissue sections were deparaffinized with xylene and rehydrated in ethanol solutions ranging from 100% to 30%. Antigen-retrieval was performed by boiling the sections in 10 mM citrate buffer pH 6.0. The sections were blocked for 1 h with 5% fetal bovine serum (FBS) in PBS. Afterwards the antibodies for TGM2 and 3 were added (1:200 diluted in PBS containing 5% FBS) and the sections incubated overnight at 4 °C in a humid chamber followed by three washing steps in PBS. Secondary antibodies coupled to the AlexaFluor647 dye (α-mouse-IgG1 for TGM2 α-rabbit-IgG for TGM3, 1:1,000 diluted in PBS containing 5% FBS) were added together with the UEA1 lectin (10 μg/ml) conjugated to the rhodamine dye for 1 h. After three washing steps in PBS the nuclei were stained with Sytox green (Thermo Fisher Scientific) for 5 min. After one additional washing step the sections were mounted using ProLong Gold-Antifade mountant (Thermo Fisher Scientific) and visualized by confocal microscopy (Zeiss Examiner 2.1; LSM 700).

**Collection of colonic mucus**. Mice were anaesthesized with isofluorane and sacrificed by cervical dislocation. The colon was collected by dissection and opened under a stereo microscope. Fecal pellets were removed using forceps and mucus was gently scraped off from the epithelial surface using a blunted spatula. Mucus that was sticking to the spatula and detached mucus from the epithelial surface was collected using a pipette and subsequently emulsified in TBS buffer (50 mM Tris pH 8.0; 150 mM NaCl) to a total volume of 200 μl. The samples were stored until further use at −80 °C.

**Transglutaminase activity assays**. Collected mucus samples were separated into the insoluble MUC2 fraction and supernatant by centrifugation (16,000 × g; 30 min; 4 °C). Mucus supernatants (30 μg) were incubated either with 10 μM of the TGM2- or TGM3-specificic glutamine-donor peptides T26[21] or E51[22] or the amine-donor compound 5-Biotinyl-pentylamine for 1 h at 37 °C. Control reactions were performed in the presence of 25 mM IAA and the reactions were stopped by the addition of SDS-loading buffer and heating to 95 °C for 5 min. Reaction products were separated by SDS-PAGE on 4–15% gradient gels followed by semidry transfer to PVDF membranes. After blocking with 3% BSA in TBS buffer the membrane was incubated with streptavidin coupled to AlexaFluor 680 (Invitrogen; 1:20,000, in TBS buffer containing 0.1% Tween20; 0.02% SDS and 3% BSA) and the incorporation of substrates detected on a LI-COR Odyssey Clx workstation. The Western Blots were analyzed using the Image Studio lite software (version 2.1).

Quantitative determination of TGM activity was performed according to the method described by Trigwell and coworkers[45]. Briefly, maxisorb 96-well plates (Thermo) were coated with 250 μl of a 0.1% casein solution in 50 mM sodium carbonate pH 9.8 for 12 h. After emptying and washing 250 μl blocking solution (0.1% BSA in 50 mM sodium carbonate pH 9.8) was added and incubated for 1 h at 37 °C. After washing, 150 μl reaction buffer (100 mM TrisHCl pH 8.5, 6.7 mM CaCl₂, 13.3 mM DTT containing either 10 μM biotinylated TGM-substrate peptide E51; T26, respectively or 5 μM biotinylated TGM-substrate peptide A25) for the

respective TGM standards was added to the wells. For the analysis of mucus samples, DTT and calcium were omitted from the reaction buffer. Measurements were carried out in triplicate per biological replicate. The reactions were started by the addition of either 50 μl TGM standards (0; 25; 50; 75; 100; 125 mU/well) or mucus samples and incubated for 1 h at 37 °C on a rotational shaker set to 100 rpm. Afterwards, the reactions were stopped by emptying the wells and washing. The incorporation of the substrates in the casein matrix was probed by the addition of 200 μl Extravidin solution (Extravidin-peroxidase (1:10,000 in 100 mM TrisHCl pH 8.5 containing 1% BSA) for 1 h and gentle shaking. Biotin-Extravidin binding was visualized by adding 200 μl TMB developing solution (3,3′,5,5′-Tetramethylbenzidine) and the reaction stopped by adding 50 μl 5 M H₂SO₄. The absorbance of reaction and standard wells was recorded at 450 nm on a Victor2 Wallac workstation (Perkin Elmer). The activities of the samples were subsequently normalized against the protein content of the sample using the BCA method.

**Ex vivo analysis of transglutaminase activity**. Mice were anaesthetized using isofluorane and sacrificed by cervical dislocation. The colon was collected by dissection and flushed for the removal of intestinal content using Krebs buffer as previously described[46]. After removal of the muscle layer by microdissection the tissue was mounted in an in-house built horizontal chamber allowing basolateral perfusion with Krebs-Glucose buffer and apical Krebs-mannitol buffer (Fig. 2e). 2 μM FITC-labeled E51 probe in Krebs-mannitol buffer was added and the tissue incubated for 30 min at 37 °C. Afterwards, non-incorporated probe molecules were washed away with Krebs-mannitol buffer followed by analysis of incorporation of the TGM3-substrate peptide on an upright LSM700 confocal microscope (Zeiss Examiner 2.1) equipped with a ×20 immersion lens (Pan-Apochromat ×20/1.0 DIC 75 mm; Carl Zeiss, Germany). Images were acquired using Zen Black software (Carl Zeiss) and z-stacks were exported to TIFF format using Imaris software. Inhibition of transglutaminase activity in WT mice was achieved by adding 5 μM Z-DON[23] (Zedira) together with the TGM3 substrate.

**Ex vivo mucus integrity assay**. Tissue was collected as described for the ex vivo analysis of transglutaminase activity. Following mounting in the perfusion chamber, tissue was stained with Syto 9 (1:500 in Kreb's-mannitol buffer; Thermo Fisher) and the mucus layer was visualized by the addition of 1 μm fluorescent beads (Thermo Fisher). 20 mg/ml of pronase was added to the apical Krebs-mannitol buffer and the integrity of the mucus layer was monitored on an upright LSM900 confocal microscope (Carl Zeiss) using a water Pan-Apochromat ×20/1.0 DIC 75 mm lens (Carl Zeiss; Germany). Tissue explants were maintained at 37 °C throughout the experiments. Briefly, z-stacks were acquired every 5 min (total time 1 h) using Zen Blue software (version 3.1; Carl Zeiss, Germany). In order to monitor mucus integrity beads and tissue surfaces were mapped to isosurfaces using Imaris software as described previously[47], data regarding the position of the fluorescent beads in relation to the tissue surface over time was then extracted and analyzed to generate normalized positional data over time (Prism version 9.1.0, Graphpad).

**Colitis induction by DSS**. Six- to eight-week-old female wild-type (WT) C57/BL6 and $Tgm3^{-/-}$ mice were cohoused for 4 to 5 weeks. Colitis was induced by adding 3% (w/v) dextran sodium sulfate (DSS) to the drinking water. Mice could drink *ad libitum*. The mice were sacrificed after eight days or if their body weight dropped by 10% from the initial weight. The probability of survival was defined when mice died or if they showed a body weight loss >10%. The colon was dissected and its length measured from cecum to anus and subsequently normalized against the initial body weight of the respective animal. Afterwards, the colon was flushed with PBS for the removal of fecal content. The colons were fixed as Swiss rolls in 4% paraformaldehyde and stained for hematoxylin/eosin and Alcian Blue-PAS. The disease activity index (DAI) was calculated as the sum of the combined scores for stool consistency, hematochezia and weight loss according to the methods of Friedman and coworkers[48]. The detection of occult blood was performed using the Hemocccult Guaiac Fecal Occult Blood Test kit (Beckman Coulter) according to the manufacturer's instructions. Two litters of each mouse strain with five animals per litter were analyzed.

**Composite agarose-PAGE**. The separation of MUC2 was performed according to the protocol of Schulz and coworkers[49]. Briefly, mucus was scraped from mouse colon and emulsified in TBS. Mucus/Muc2 was precipitated by centrifugation at 16,000 × g and 4 °C for 30 min. The mucus was solubilized by the addition of reducing gel-loading buffer (62.5 mM TrisHCl pH 6.8, 2% SDS, 50 mM DTT 20% (v/v) glycerol). 67 μg were separated via AgPAGE for 3.5 h at 30 mA. The gels were either stained with Alcian Blue or MUC2 was detected by in-gel immunodetection. Alcian Blue-stained gels were scanned on a GelDoc EZ imager using the image lab™ software (BioRad). For in-gel immunodetection, the gels were fixed in 50% (v/v) 2-Propanol/ 5% (v/v) acetic acid for 15 min and gentle shaking followed by 30 min washing in water. The primary antibody against MUC2 (Genetex; 1:500) was added for 12 h at 4 °C in PBS-T buffer containing 5% BSA. After three washing steps with PBS-T for 10 min each, the secondary antibody α-rabbit-IgG-Licor790 (LiCOR, 1:5000) was added for 1 h at ambient temperature. After three to five

extensive additional washing steps, the immunostained gel was scanned with a LiCOR Odyssey Clx instrument and analyzed using the Image Studio lite software (LiCOR).

**Thermofluor assay.** Mucus from the indicated mouse strains was scraped from their distal colons and emulsified in TBS buffer. Insoluble mucins were washed twice in TBS and recovered by centrifugation (16,000 × $g$; 4 °C; 30 min). The protein concentration of the supernatant was determined and the mucus pellet emulsified to a concentration of 1 mg/ml in each sample. 45 µl of sample or TBS control were mixed with five µl of a 200-fold stock solution of SyproOrange (Molecular Probes) and subjected to an increasing temperature gradient of 0.5 °C every 30 s from 25 to 99 °C in a CFX96 Real-time system (BioRad). The fluorescence was recorded every 30 s and the fluorescence intensity of the TBS control subtracted. To rescue the properties of mucus from WT mice 1 U recombinant TGM3 and 4 mM CaCl$_2$ were applied to the mucus from $Tgm3^{-/-}$ mice and incubated for 1 h at 37 °C. The reaction was terminated by the addition of 5 mM IAA. The buffer controls for this part of the experiment were treated accordingly and the melting curve recorded as described above. Three animals per strain were analyzed in technical triplicates.

**Analysis of MUC2 depolymerization by turbidity measurement.** The turbidity of the supernatant from scraped mucus samples was recorded at a wavelength of 600 nm in a Spectramax photometer after precipitation of insoluble mucus by centrifugation (1000 × $g$, 30 min, 4 °C) followed by subsequent normalization against the protein concentration of the respective sample. The mean of the normalized absorbance from WT samples was set to 100% and used as reference point for all samples analyzed. Three animals per mouse strain were analyzed.

**Single cell transcriptomic analysis.** RNA-seq data were extracted from a recently published study[20]. Briefly, goblet cells and non-goblet cells from the RedMUC2 reporter mouse strain were isolated by FACS as described recently[20]. The used bulk RNA-seq data (GSE144363) are deposited in GEO and belong to the superserie GSE144436. The quality of the data was assessed with FastQC (version 0.11.2) and filtered using Prinseq (version 0.20.3). The reads were aligned against the mouse reference genome mm10 with STAR (version 2.5.2b) and the number of mapped reads was calculated with HTseq (version 0.6.1p1). Data normalization, differential expression and statistical analysis were made with DESeq2 (version 1.14) in R.

**In-gel digestion and mass spectrometric analyses.** Briefly, protein bands of interest were excised from the gel and washed with 50% acetonitrile and dried in a vacuum centrifuge followed by reduction with 10 mM DTT for 30 min at 56 °C and subsequent alkylation with 55 mM IAA for 30 min at room temperature. Trypsin was added at a ratio of 1:50 and the samples incubated for 12 h at 37 °C. Afterwards, AspN was added at a ratio of 1:50 and the samples incubated for additional 5 h at 37 °C. The digestion was stopped by adding TFA to a concentration of 0.5%.

Salt and buffer components were removed by in-house stage tips equipped with C18 resin[50] and the peptides dissolved in 0.1% formic acid. The samples were analyzed on a Q-Exactive mass spectrometer as described earlier[51].

**MS data analysis.** MS raw files were transformed into *.mgf files using the MS convert software. These files were analyzed using the MASCOT search engine (Matrix Science). Searches were performed against the UniProt database (version 06/2017 containing 554515 sequences) and an in-house database (http://www.medkem.gu.se/mucinbiology/databases/index.html) containing all human and mouse mucin sequences. Searches were performed with the following parameters: mass tolerance for the precursor ion of 5 ppm; tolerance for fragment ions 0.2 Da; full specificity for trypsin/AspN with a maximum of two missed cleavages; carbamidomethylation as static modification and oxidation of methionine as variable modification. The sequence coverage of individual MUC2 domains was obtained by expressing the number of identified amino acids as a percentage of the total number of amino acids in a given domain. Amino acids that were identified in overlapping peptides were only counted once. Only residues from peptides with an ion score >25 were taken into account.

TGM-catalyzed cross-linked peptides were searched using the StavroX software tool (version 3.6.6)[52] against theoretical intra- and intermolecular isopeptide cross-linked (di)peptides of the murine MUC2 (UniProt identifier: Q80Z19) using the following parameters: mass tolerance for the precursor ion of 2 ppm; tolerance for fragment ions 20 ppm; full specificity for trypsin/AspN with a maximum of three missed cleavages; Gln and Lys as cross-linking sites; composition of the cross-link –NH$_3$; carbamidomethylation as static modification and methionine oxidation as variable modification. Label-free mass spectrometric quantification of TGM isozymes was extracted from the study by Nystrom et al.[20].

**Statistical analysis.** Statistical analyses were performed using the Prism software (version 9.0.1; GraphPad). Turbidity, body weight and colon length were compared using the unpaired t-test with Welch's correction. DAI scores were compared by multiple unpaired t-tests using the Holm-Sidák correction. The probalitiy of survival was compared using the Wilcoxon test. Significance was accepted when p values were below 0.05. Data are expressed as mean ± standard deviation.

**Reporting summary.** Further information on research design is available in the Nature Research Reporting Summary linked to this article.

## Data availability

All data are available within the article and Supplementary data files, or available from the corresponding author upon reasonable request. Source data are provided with this paper. The proteomics dataset for label-free quantification used has been published[20] and deposited to the ProteomeXchange Consortium (http://proteomecentral.proteomexchange.org) with the dataset identifier PXD011527. The mass spectrometry data for the analyses of MUC2 monomers generated in this study have been deposited to the ProteomeXchange Consortium with the dataset identifier PXD029071 (http://proteomecentral.proteomexchange.org). The MASCOT (http://matrixscience.org) and StavroX (https://stavrox.com/) result files are included in the Supplementary data files. The bulk RNA-seq data (GSE144363) are deposited in GEO (https://www.ncbi.nlm.nih.gov/geo/) and belong to the superserie GSE144436 and have been published[20]. Source data are provided with this paper.

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

## Acknowledgements

The authors acknowledge Ludvig Sollid, University of Oslo and Eleonara Candi, University of Rome for providing the *Tgm2*$^{-/-}$ and *Tgm3*$^{-/-}$ mice strains. This work was supported by the European Research Council ERC (694181) GCH, National Institute of Allergy and Infectious Diseases (U01AI095473, the content is solely the responsibility of the authors and does not necessarily represent the official views of the NIH) GCH, The Knut and Alice Wallenberg Foundation (2017.0028) GCH, Swedish Research Council (2017-00958) GCH, The Swedish Cancer Foundation (CAN 2017/360) GCH, IngaBritt and Arne Lundberg Foundation (2018-0117) GCH, Sahlgren's University Hospital (ALFGBG-440741, The ALF agreement 236501) GCH, Bill and Melinda Gates Foundation (OPP1202459) GCH, Wilhelm and Martina Lundgren's Foundation GCH.

## Author contributions

J.D.A.S. performed experiments and analyzed data; B.D. performed experiments and analyzed data; E.E.L.N. performed experiments and analyzed data, L.A. performed experiments and analyzed data; G.M.H.B. performed experiments and analyzed data; B.M.A. performed experiments and analyzed data; M.E.V.J. data analysis; G.C.H. conceptualized the study, analyzed data; C.V.R. conceptualized the study, performed experiments and analyzed data. G.C.H. and C.V.R. wrote the paper. All authors reviewed the paper and accepted the final version.

## Funding

## Competing interests

The authors declare no competing interests.
