## [Peer Review File · Nature Communications]

Reviewers' Comments:

Reviewer #1:

Remarks to the Author:

The manuscript "Transglutaminase 3 crosslinks secreted MUC2 and stabilizes the colonic mucus layer" by Sharpen et al seeks to understand the basis of how the Muc2-mucin is able to form a stable protective polymeric structure when exposed to a vast proteolytically active luminal environment. Given the importance of this mucus system to gut homeostasis, as evidenced by its defects in inflammatory bowel disease and colon cancer patients, and spontaneous colitis and cancer in mice lacking Muc2, the rationale for this study is sound and this study is potentially of great import for our understanding of mechanisms of mucus and gut homeostasis and novel therapeutic targets for chronic diseases above. To this end, Sharpen et al have claimed to have identified Transglutaminase 3 (TGM), a member of the transglutaminase family that is known to crosslink glutamine and lysine residues to form intra- and intermolecular isopeptide bonds that can be resistant to proteolytic attack, as an important factor to stabilize the Muc2 polymer. Insightfully, it is proposed that TGM3 is not only expressed in Goblet cells, but its activity happens within the secreted mucus layer in the lumen. This is a fascinating concept that boosts our understanding of how the vital mucus layer forms an effective barrier. Using a series of complex but elegant biochemical, ex vivo imaging, and in vivo disease modeling approaches derived from WT and TGM2 and TGM3 KO mice, Sharpen et al provide some compelling evidence for the role of TGM3 in regulating luminal Muc2 crosslinking. However, there are a few notable pieces of evidence that would be helpful to strengthen their conclusions.

MAJOR ISSUES

1. A major claim by the authors is that the TGM3-mediated Muc2 cross-linking renders the mucus barrier layer more resistant to attack by the plethora of luminal microbially or host-derived proteases. This concept was beautifully supported by the ex-vivo studies in Fig 3h and i. However, as it stands, the authors point to the physiologic relevance of this phenotype by correlating the biochemical and ex-vivo studies with the susceptibility of TGM3 KO mice to DSS. But this is quite a jump to conclusions, as it presumes no other function of TGM3 could account for the DSS susceptibility. What is missing is in-situ data of the mucus layer-microbe interactions in the WT vs TGM3 KO or TGM2 KO littermates, for example by Carnoys-fixed sections, to directly investigate the link between TGM activity, mucus function in vivo/in situ, and DSS susceptibility. If the hypothesis is true, we might expect to see some defect in the mucus structure or barrier properties (i.e. increased bacterial penetrance) in the TGM3^{-/-} mucus by Muc2/FISH labeling. The data in Fig 1 c&d in fact seems to support this as the mucus seems disrupted in the Tgm3^{-/-} mice. If no difference is seen, that does not negate the conclusions of the authors as this technique may not be sensitive enough to detect this. However, it should still be done to link the ex vivo/biochemical data with the in vivo physiology.

2. The dominant role of TGM3 vs TGM2 is convincing as currently presented, with the data from the knockout lines particularly compelling. The authors ascribe most of the activity of TGM3 coming from goblet cells which makes sense since it should in theory be co-secreted with Muc2. However, the IF and scRNAseq data (Fig 1) show it's also abundantly in non-goblet epithelial cells. Is it possible that neighboring epithelial cells are also releasing TGM3 into the lumen and this is a co-operative effort of non-goblet cells to help stabilize the mucus? This might be best addressed in the Discussion.

3. The hydrophobicity study in Fig.3g is intriguing, and I've never seen that done before. However, it's unclear how that data is contributing to the conclusions of this study. What is the ultimate interpretation of this data?

4. The authors make the point that there are no intrinsic defects in mucus formation in the TGM3 KO mice (line, as shown in Fig S3). This makes sense since the mucus is not yet exposed to the luminal mucin-degrading enzymes. To gain some insight into what luminal components are

contributing to the faster-degraded mucus, it may be helpful to give TGM3 KO mice a dose of broad-spectrum antibiotics over a few days (not too long or that may impair the mucus production) to clear the microbes and then assess the presence of the faster migrating band in the TGM3 KO mice. This would allow dissection of whether the microbiota is responsible for the degraded mucus in TGM3 KO mice. If no impact is seen, that would argue for a role of host proteases.

5. Several studies are done ostensibly with purified mucus (i.e. Fig. 2a and corresponding results), but it is not clear in the Methods how this mucus was acquired. Was it extracted and purified according to normal protocols (Extraction in GuCl, Reduction in DTT, followed by Alkylation and dialysis?).

6.

MINOR ISSUES

1. In the Discussion (Lines 353-354), the authors speculate that several factors, including O-glycosylation, can contribute to resistance of Muc2 to degradation. There is a study that has directly addressed this that co-authors MEV Johansson and GC Hansson are on (PMID: 27143302) – this should be referenced here.

2. Line 351-53 ("Thus, ...") - this sentence is not clear.

3. Line 35: The statement that spontaneous colitis is a "pre-stage to colon carcinoma" is a bit oversimplified. It is true that chronic colitis can increase risk of a particular subtype of colon cancer – colitis-associated cancer--, but many other colon cancers are not associated with chronic colitis per se, but instead somatic or inherited mutations in oncogenes/tumor suppressors. Since this paper has little to do with colon cancer, it is suggested to leave this statement out and focus on the chronic colitis, IBD aspect. If this is important to the authors however to include the reference to colon cancer, then it is suggested to at least change "Pre-stage" to "Predispose to" or "increase risk for" "some types of" colon carcinoma.

Reviewer #2:

Remarks to the Author:

The authors report the first extensive investigation of transglutaminase cross-linking in the colonic mucus layer, revealing that TG3 is major isoform whose activity is biologically relevant. Their results very clearly support the role of transglutaminase 3 (TGM3) as being intrinsically active in cross-linking native acyl-donor and -acceptor protein substrates. They also demonstrate the biochemical importance of this cross-linking activity and the biological consequences of the lack of this activity.

Experimentally, they first tested the activity of different isozymes and concluded TG3 was active. mRNA levels showed only TGM2 and TGM3 were present. Protein abundance was evaluated by MS. Detection of higher mass forms that were identified as TGM3 by immunoblotting suggested self-multimerization.

The authors used peptide sequences specific to TGM2 and TGM3 to clearly distinguish their intrinsic activities in the colonic mucus.

One of their experiments relied on the use of Z-DON to block TGase activity, but Z-DON was designed as an inhibitor of TGM2. Is there independent evidence that it blocks the activity of TGM3? If so, they should add a reference to this effect. If not, they should demonstrate this through in vitro inhibition kinetic experiments. They should also report where they obtained Z-DON, presumably from Zedira.

Additional experiments also characterized the biochemical properties of the mucus barrier in Tgm3^{-/-} tissue. Importantly, they showed that the TGM3-mediated cross-linking activity slows degradation of mucous component MUC2. Also, TG3^{-/-} mice showed susceptibility to colitis.

This is very important work and a very thorough investigation of an exciting discovery.

The manuscript is well-written, and very clear apart from one paragraph:

p. 6, line 141 – The authors say they used 5-BP in the place of the Gln-donor substrate (T26 or E51) “as acyl-donor”. This is incorrect, but I think it is just a typo. 5-BP cannot be an acyl-donor, but rather it is an “acyl-acceptor”. The confusion continues in the next sentence. I think it should read “similar to the results from the acyl-donor experiments” (assuming they are referring to biotinylated T26 and E51) “specific signals were detected when the acyl-acceptor compound” (assuming this is 5-BP) “was added...”

Also, by way of another minor suggestion, on p. 6, line 145, and also p. 13 line 314: I think the word “substrates” would be convey more significance than simply “molecules”

REVIEWER COMMENTS

Reviewer #1 (Remarks to the Author):

The manuscript "Transglutaminase 3 crosslinks secreted MUC2 and stabilizes the colonic mucus layer" by Sharpen et al seeks to understand the basis of how the Muc2-mucin is able to form a stable protective polymeric structure when exposed to a vast proteolytically active luminal environment. Given the importance of this mucus system to gut homeostasis, as evidenced by its defects in inflammatory bowel disease and colon cancer patients, and spontaneous colitis and cancer in mice lacking Muc2, the rationale for this study is sound and this study is potentially of great import for our understanding of mechanisms of mucus and gut homeostasis and novel therapeutic targets for chronic diseases above. To this end, Sharpen et al have claimed to have identified Transglutaminase 3 (TGM), a member of the transglutaminase family that is known to crosslink glutamine and lysine residues to form intra- and intermolecular isopeptide bonds that can be resistant to proteolytic attack, as an important factor to stabilize the Muc2 polymer. Insightfully, it is proposed that TGM3 is not only expressed in Goblet cells, but its activity happens within the secreted mucus layer in the lumen. This is a fascinating concept that boosts our understanding of how the vital mucus layer forms an effective barrier. Using a series of complex but elegant biochemical, ex vivo imaging, and in vivo disease modeling approaches derived from WT and TGM2 and TGM3 KO mice, Sharpen et al provide some compelling evidence for the role of TGM3 in regulating luminal Muc2 crosslinking. However, there are a few notable pieces of evidence that would be helpful to strengthen their conclusions.

MAJOR ISSUES

1. A major claim by the authors is that the TGM3-mediated Muc2 cross-linking renders the mucus barrier layer more resistant to attack by the plethora of luminal microbially or host-derived proteases. This concept was beautifully supported by the ex-vivo studies in Fig 3h and i. However, as it stands, the authors point to the physiologic relevance of this phenotype by correlating the biochemical and ex-vivo studies with the susceptibility of TGM3 KO mice to DSS. But this is quite a jump to conclusions, as it presumes no other function of TGM3 could account for the DSS susceptibility. What is missing is in-situ data of the mucus layer-microbe interactions in the WT vs TGM3 KO or TGM2 KO littermates, for example by Carnoys-fixed sections, to directly investigate the link between TGM activity, mucus function in vivo/in situ, and DSS susceptibility. If the hypothesis is true, we might expect to see some defect in the mucus structure or barrier properties (i.e. increased bacterial penetrance) in the TGM3^{-/-} mucus by Muc2/FISH labeling. The data in Fig 1 c&d in fact seems to support this as the mucus seems disrupted in the Tgm3^{-/-} mice. If no difference is seen, that does not negate the conclusions of the authors as this technique may not be sensitive enough to detect this. However, it should still be done to link the ex vivo/biochemical data with the in vivo physiology.

Comment of the authors:

The authors thank the reviewer for the suggested experiment. We have now addressed this point by performing a combined FISH/MUC2 staining. Whereas tissue sections from WT animals showed a stratified mucus organization the mucus of *Tgm3*^{-/-} mice had a more disrupted appearance and seemed to be more detached from the epithelium. Furthermore, the separation distance between bacteria and the tissue surface was decreased and the microorganisms were in some parts in close contact with the epithelium. These data are now included in the manuscript as Suppl. Fig. S5. The new data support the conclusion that the mucus layer is compromised in the knock out animals and suggests an additional

explanation for the early onset of colitis in these mice.

2. The dominant role of TGM3 vs TGM2 is convincing as currently presented, with the data from the knockout lines particularly compelling. The authors ascribe most of the activity of TGM3 coming from goblet cells which makes sense since it should in theory be co-secreted with Muc2. However, the IF and scRNAseq data (Fig 1) show its also abundantly in non-goblet epithelial cells. Is it possible that neighboring epithelial cells are also releasing TGM3 into the lumen and this is a co-operative effort of non-goblet cells to help stabilize the mucus? This might be best addressed in the Discussion.

Comment of the authors:

The authors agree that it is possible that the neighboring non-goblet cells might also contribute to the TGM3-mediated stabilizing effect of the mucus. The authors addressed this point by including the sentence (after line 355) in the discussion section: “In addition to goblet cells also neighboring enterocytes could contribute to the TGM3-mediated mucus stabilization after shedding and subsequent release of their cellular content as this acyl-transferase was also abundant in these cells.”

3. The hydrophobicity study in Fig.3g is intriguing, and I've never seen that done before. However, it's unclear how that data is contributing to the conclusions of this study. What is the ultimate interpretation of this data?

Comment of the authors:

Our hydrophobicity studies aim to indirectly show the presence of isopeptide-cross-links as the formation of such covalent bonds prevents proteins from a normal temperature-induced unfolding as it has been shown for bacterial pili proteins (Ref 38 and 39). Therefore, the decreased fluorescence gain of WT-MUC2 compared to *Tgm3*^{-/-}-MUC2 during the heat-induced denaturation suggests the presence of these natural cross-links. Furthermore, the hydrophobic behaviour of MUC2 from *Tgm3*^{-/-} animals is partly rescued after treatment with recombinantly expressed TGM3 thereby supporting this data interpretation.

Thus, our ultimate interpretation of this data is that the hydrophobic character of MUC2/mucins is at least partly a result of the post-translational generation of isopeptide-based cross-links catalyzed by transglutaminases as discussed in lines 384-9.

4. The authors make the point that there are no intrinsic defects in mucus formation in the TGM3 KO mice (line, as shown in Fig S3). This makes sense since the mucus is not yet exposed to the luminal mucin-degrading enzymes. To gain some insight into what luminal components are contributing to the faster-degraded mucus, it may be helpful to give TGM3 KO mice a dose of broad-spectrum antibiotics over a few days (not too long or that may impair the mucus production) to clear the microbes and then assess the presence of the faster migrating band in the TGM3 KO mice. This would allow dissection of whether the microbiota is responsible for the degraded mucus in TGM3 KO mice. If no impact is seen, that would argue for a role of host proteases.

Comment of the authors: The authors thank for the suggested experiment and addressed this issue by co-housing WT and *Tgm3*^{-/-} mice followed by the supplementation of an antibiotics cocktail in the drinking water for four days. The short antibiotics treatment led to a reduction of the number of bacteria on average by two orders of magnitude as indicated by amplifying the 16S DNA locus (Suppl. Fig. S3a). In the next step the electrophoretic migration pattern of MUC2 after the treatment was analysed and no

difference between MUC2 from WT and knock out animals detected (Suppl. Fig. S3b). This data suggests that microbial proteases are mainly responsible for the degradation of colonic mucus fitting to the known observation that host proteases are mainly active in the small intestine in order to catalyze food break down. This data are now included in the manuscript (line 217-24) and as Suppl. Fig. S3.

5. Several studies are done ostensibly with purified mucus (i.e. Fig. 2a and corresponding results), but it is not clear in the Methods how this mucus was acquired. Was it extracted and purified according to normal protocols (Extraction in GuCl, Reduction in DTT, followed by Alkylation and dialysis?).

Comment of the authors:

The authors addressed this point by a more detailed description of the respective methods sections (p. 20 line 474-481; p.23 line 567 and 24 line 581). Briefly, mucus was acquired by gentle scraping from the colonic surface and emulsification in TBS buffer. For the activity assays mucus was not reduced or alkylated (Fig. 2a-d). For the in-gel immunodetection (Fig. 3b) MUC2 was reduced by the addition of loading buffer before agarose-PAGE separation. For the analyses of its susceptibility to LysC degradation (Fig. 3c) in the different mouse strains the mucus was reduced with DTT followed by treatment in the absence or presence of the protease (Fig. 3d). Afterwards, protein bands of interest were reduced and alkylated prior to mass spectrometric analysis. The turbidity analysis for the estimation of an intact MUC2 gel network (Fig. 3e) was performed on non-reduced mucus. For the analysis of its hydrophobicity (Fig. 3g) the emulsified MUC2 molecules were washed in TBS before incubation with the hydrophobic dye SyproOrange and recording of the melting curve.

MINOR ISSUES

1. In the Discussion (Lines 353-354), the authors speculate that several factors, including O-glycosylation, can contribute to resistance of Muc2 to degradation. There is a study that has directly addressed this that co-authors MEV Johansson and GC Hansson are on (PMID: 27143302) – this should be referenced here.

Comment of the authors: The authors agree and addressed this point by including this publication in the respective part of the discussion section in line 377 as reference 37.

2. Line 351-53 (“Thus, …”) - this sentence is not clear.

Comment of the authors: The authors addressed this point by changing
“ Thus, leaving a central part of the MUC2 mucin consisting of the two highly glycosylated PTS sequences connected via the second CysD domain behind. “

To

“Thus, leaving the central part of the MUC2 mucin consisting of the two highly glycosylated PTS sequences linked via the CysD2 domain that is located between them, intact (Fig. 3a).” (line 373-5).

3. Line 35: The statement that spontaneous colitis is a “pre-stage to colon carcinoma” is a bit oversimplified. It is true that chronic colitis can increase risk of a particular subtype of colon cancer – colitis-associated cancer--, but many other colon cancers are not associated with chronic colitis per se, but instead somatic or inherited mutations in oncogenes/tumor suppressors. Since this paper has little to do with colon cancer, it is suggested to leave this statement out and focus on the chronic colitis, IBD aspect. If this is important to the authors however to include the reference to colon cancer, then it is suggested to at least change “Pre-stage” to “Predispose to” or “increase risk for” “some types of” colon carcinoma.

Comment of the authors: The authors addressed this point by changing “...pre-stage to colon carcinoma...” to “, leading to an increased risk for some types of colon carcinoma” (line 20-21).

Reviewer #2 (Remarks to the Author):

The authors report the first extensive investigation of transglutaminase cross-linking in the colonic mucus layer, revealing that TG3 is major isoform whose activity is biologically relevant. Their results very clearly support the role of transglutaminase 3 (TGM3) as being intrinsically active in cross-linking native acyl-donor and -acceptor protein substrates. They also demonstrate the biochemical importance of this cross-linking activity and the biological consequences of the lack of this activity.

Experimentally, they first tested the activity of different isozymes and concluded TG3 was active. mRNA levels showed only TGM2 and TGM3 were present. Protein abundance was evaluated by MS. Detection of higher mass forms that were identified as TGM3 by immunoblotting suggested self-multimerization.

The authors used peptide sequences specific to TGM2 and TGM3 to clearly distinguish their intrinsic activities in the colonic mucus.

One of their experiments relied on the use of Z-DON to block TGase activity, but Z-DON was designed as an inhibitor of TGM2. Is there independent evidence that it blocks the activity of TGM3? If so, they should add a reference to this effect. If not, they should demonstrate this through in vitro inhibition kinetic experiments. They should also report where they obtained Z-DON, presumably from Zedira.

Comment of the authors: The authors addressed this point by including the study from Schaertl *et al.* (PMID: 20395409) as reference 23 in the manuscript (p. 7 line 175 and p. 19 line 455).

This work has extensively analysed the inhibitory effect of Z-DON on TGM1, 2, 3, 6 and FXIIIa. Z-DON showed an IC50 value of 0.2 μ M towards TGM3. We ordered Z-DON from Zedira and added the compound to the list of chemicals.

Additional experiments also characterized the biochemical properties of the mucus barrier in Tgm3^{-/-} tissue. Importantly, they showed that the TGM3-mediated cross-linking activity slows degradation of mucous component MUC2. Also, TG3^{-/-} mice showed susceptibility to colitis.

This is very important work and a very thorough investigation of an exciting discovery.

The manuscript is well-written, and very clear apart from one paragraph:

p. 6, line 141 – The authors say they used 5-BP in the place of the Gln-donor substrate (T26 or E51) “as acyl-donor”. This is incorrect, but I think it is just a typo. 5-BP cannot be an acyl-donor, but rather it is an “acyl-acceptor”. The confusion continues in the next sentence. I think it should read “similar to the results from the acyl-donor experiments” (assuming they are referring to biotinylated T26 and E51) “specific signals were detected when the acyl-acceptor compound” (assuming this is 5-BP) “was added...”

Comment of the authors: The authors agree and apologize for this mixing up of terms and corrected the terms in the respective parts of the revised version of the manuscript (page 6).

Also, by way of another minor suggestion, on p. 6, line 145, and also p. 13 line 314: I think the word “substrates” would be convey more significance than simply “molecules”

Comment of the authors: The authors thank for this suggestion and changed molecules to substrates in the manuscript.

Reviewers' Comments:

Reviewer #1:

Remarks to the Author:

The authors have adequately addressed my concerns raised in the previous submission. The revised version is significantly improved over the original and makes a very strong case for the novel role of Tgm3 in maintaining mucus integrity and homeostasis in the mammalian intestinal tract.

Reviewer #2:

Remarks to the Author:

The revised version of this manuscript addresses all of the corrections and concerns that I provided during the initial review.

Furthermore, in my opinion that authors have also done well to respond to the comments of the other Reviewer; however, my opinion is not as relevant in this regard as the other Reviewer's. :)

To re-iterate my previous summary, I think this is very important work that provides fascinating insight into the physiological role of TG3. The authors' data support their conclusions and the manuscript is well written. I recommend publication.